# Artichoke and Bergamot Phytosome Alliance: A Randomized Double Blind Clinical Trial in Mild Hypercholesterolemia

**DOI:** 10.3390/nu14010108

**Published:** 2021-12-27

**Authors:** Antonella Riva, Giovanna Petrangolini, Pietro Allegrini, Simone Perna, Attilio Giacosa, Gabriella Peroni, Milena Anna Faliva, Maurizio Naso, Mariangela Rondanelli

**Affiliations:** 1Development Department, Indena SpA, 20139 Milan, Italy; antonella.riva@indena.com (A.R.); giovanna.petrangolini@indena.com (G.P.); pietro.allegrini@indena.com (P.A.); 2Department of Biology, College of Science, Sakhir Campus, University of Bahrain, Zallaq 32038, Bahrain; simoneperna@hotmail.it; 3CDI—Centro Diagnostico Italiano, 20147 Milan, Italy; attilio.giacosa@gmail.com; 4Endocrinology and Nutrition Unit, Azienda di Servizi alla Persona ‘‘Istituto Santa Margherita’’, University of Pavia, 27100 Pavia, Italy; milena.faliva@gmail.com (M.A.F.); mau.na.mn@gmail.com (M.N.); 5IRCCS Mondino Foundation, 27100 Pavia, Italy; mariangela.rondanelli@unipv.it; 6Unit of Human and Clinical Nutrition, Department of Public Health, Experimental and Forensic Medicine, University of Pavia, 27100 Pavia, Italy

**Keywords:** bergamot, artichoke, poor responders, body composition, mild hypercholesterolemia, phytosome^®^

## Abstract

Botanicals are natural alternatives to pharmacological therapies that aim at reducing hypercholesterolemia. In this context, despite bergamot being effective in modulating lipid profile, some subjects failed to achieve a satisfactory response to supplementation. The aim of this study was to evaluate whether the association of 600 mg of bergamot phytosome^®^ (from *Citrus Bergamia* Risso) and 100 mg of artichoke leaf standardized dry extract (from *Cynara cardunculus* L.) can be an alternative in patients with mild hypercholesterolemia who are poor responders to bergamot in a 2-month randomized placebo-controlled trial. Sixty overweight adults were randomized into two groups: 30 were supplemented and 30 received a placebo. The metabolic parameters and DXA body composition were evaluated at the start, after 30 and 60 days. Between the two groups, total and LDL cholesterol in the supplemented group (compared to placebo) showed significant decreases overtime. A significant reduction of waist circumference and visceral adipose tissue (VAT) was recorded in the supplemented group (compared to placebo), even in subjects who did not follow a low-calorie diet. In conclusion, the synergism between *Citrus Bergamia* polyphenols and *Cynara cardunculus* extracts may be an effective option and may potentially broaden the therapeutic role of botanicals in dyslipidemic patients.

## 1. Introduction

According to WHO data, about 50% of deaths each year are caused by cardiovascular disease. The milestone drug therapy for cardiovascular prevention is represented by statins. These drugs are very effective in reducing serum cholesterol, but often show critical side effects [1] that lead patients to interrupt the therapy or not to have a good adherence to it [1,2,3,4,5]. Moreover, the improvement in lipid profiles is usually underachieved, as almost half of the patients undergoing statins will show a suboptimal response to these drugs [6,7].

In this panorama, use of dietary supplements is under evaluation because most of these products are almost devoid of side effects and are well tolerated in long-term therapies [2,8,9,10].

However, even in the dietary supplements scenario, it has been demonstrated that genetic factors could influence the therapeutic response. For example, the transcription factor 7-like 2 (TCF7L2)-rs7903146 polymorphism is associated with increased risk of type 2 diabetes, and the response of insulin and insulin resistance to artichoke leaf extract (ALE) may be affected by this polymorphism [11,12]. Moreover, variability in TG response has been observed after omega-3 supplementation, which may be partly explained by gene expression differences [13].

As reported by previous studies, although bergamot is effective in modulating the serum lipid profile, some subjects fail to respond to the supplementation [14,15]. We can therefore consider the presence of subjects classifiable as poor responders, even if the reasons have not yet been clarified. In fact, a study by Mollace et al. showed that 13% and 7.2% of subjects treated with a daily dose of 500 mg of bergamot extract did not modify total cholesterol and LDL cholesterol, respectively [15]. For those receiving 1000 mg/day of bergamot extract, the percentage of poor responders was reduced [15].

Moreover, human populations may respond differently to therapies and dietary supplementations, because of various factors usually not examined in preclinical studies, such as the composition of the background diet, the gut microbiota and the genetic/epigenetic variants [16].

Therefore, studying those who do not respond to therapies and dietary supplements must be a priority for scientific research in the field of dyslipidemias in order to propose a solution. For example, for dietary supplements, the synergism between different extracts could be a valid answer.

Given this background, the aim of this clinical study is to evaluate the potential greater effectiveness of a formulation consisting of the association of bergamot phytosome^®^ and artichoke leaf standardized dry extract as an hypocholesterolemic treatment in bergamot poor responders with mild hypercholesterolemia.

Bergamot (*Citrus Bergamia* Risso) extract, thanks to the presence of peculiar polyphenols, is able to counteract PCSK9, and therefore can be used both alone and in synergy with statins in order to obtain lipid lowering effects [8,10,17]. Bergamot is a citrus fruit cultivated—there are only three cultivars: Castagnaro, Fantastico and Femminello—almost exclusively in the coastal strip of Reggio Calabria (Italy) [17,18]. Furthermore, the application of phytosome technology allows for an improved bioavailability of this important class of botanical products.

The *Cynara cardunculus* L. leaf extract has demonstrated good anti-glucosidase, anti-glycation and anti-hyperglycemic effects [19]. The synergic lowering effect of these two botanical extracts on serum lipids and glucose would also contribute toward reducing inflammation and cardiovascular risk [10,20,21].

In humans, bergamot is effective in modulating total cholesterol, HDL, LDL, triglycerides and glucose levels, through antioxidant, hypoglycemic and hypolipidemic action when administered alone [15,20,21,22], or in combination with other nutraceuticals [14]. On the other hand, the combination of bergamot extract with other natural extracts appears to be effective, as shown in the paper by Cicero et al., in which bergamot extract was found to synergize with an artichoke extract and phytosterols [14]. Moreover, the synergistic activity of the bergamot extract together with the Cynara extract was effective not only on lipid metabolism, but also on oxidative stress and vascular inflammation in subjects with non-alcoholic fatty liver disease [23].

Based on these considerations, the formulation tested in this study is composed by the combination of bergamot phytosome^®^ (from *Citrus Bergamia* Risso) [24,25] and artichoke leaf extract (from *Cynara cardunculus* L.) [9]. Rationally, the association could ensure a wider range of bioavailable natural compounds with complementary mechanisms of action in dyslipidemic disorders. Potentially, this approach could allow an increase of the dyslipidemic population that could be treated with botanicals, including bergamot poor responders.

## 2. Materials and Methods

### 2.1. Experimental Design

Our study was a randomized (1:1), double blinded, placebo-controlled, parallel-group, 2-month clinical intervention study (from January 2020 to September 2021). Participants were randomized to either the supplement based on dry extract from artichoke leaf and bergamot phospholipid, or the placebo arm. Allocation to the intervention groups occurred via a computer-generated random blocks randomization list (varying block sizes) provided by an external randomization service. One researcher screened the participants enrolled and the random allocation sequence was implemented via sequentially numbered but otherwise identical sealed tablet containers allocated to participants in the order of enrollment. The randomization code was provided in sealed envelopes only to be broken at the end of the clinical trial or in the case of serious adverse events.

Sixty subjects of both sexes aged between 18 and 65 years with mild hypercholesterolemia (220–280 mg/dL), no history of cardiovascular disease (CVD), and a BMI ranging from 25 to 35 Kg/m^2^, were recruited (Figure 1). No withdrawal occurred during the trial.

### 2.2. Population

The subjects were recruited from the Dietetic and Metabolic Unit of the Santa Margherita Institute, University of Pavia, Italy. The subjects, aged 18–65 years, with a body mass index (BMI) ranging from 25 to 35 kg/m^2^, mild hypercholesterolemia (220–280 mg/dL) [26], and no history of cardiovascular disease (CVD), volunteered for the trial. The subjects were not taking any medication likely to affect lipid metabolism (such as statins). They were free of overt liver, renal and thyroid diseases. Poor responders were subjects who in a previous proof-of-concept study with bergamot phytosome (internal data) showed non-optimal decrease of cholesterol (10 mg/dL). The extent of the reduction in total cholesterol and the time of intake was decided on the basis of our previous study [27]. The experimental protocol was approved by the Ethics Committee of the University of Pavia (ethical code number: 0921/22052019) and registered at Clinicaltrials.gov (NCT04697121). All volunteers gave their written informed consent before their participation in the study. The study was conducted in accordance with the Principles of Good Clinical Practice and the Declaration of Helsinki.

### 2.3. Dietary Supplement

For the clinical study, 600 mg of bergamot (*Cytrus bergamia* polyphenols) phytosome^®^ and 100 mg of artichoke (*Cynara cardunculus* L.) leaf standardized dry extract [15] were formulated by Indena S.p.A. (Milan, Italy) into film-coated tablets containing the following ingredients: dicalcium phosphate dihydrate (Di-Cafos^®^ D160, Budenheim, Germany), magnesium carbonate (Dr. Paul Lohmann GMBH, Emmerthal, Germany), polyvinylpolypyrrolidone (PolyplasdoneTM XL, Ashland Special Ingredients, Wilmington, DE, USA), sodium croscarmellose (Solutab^®^ A-IP, Blanver Farmoquimica Ltd., Taboão da Serra, Brazil), silicon dioxide (Syloid^®^ 244FP, Grace GMBHGmbH, Worms, Germany), talc (Microtalc Pharma 50, Mondo Minerals B.VBV, Amsterdam, The Netherlands), and magnesium stearate (Ligafood^®^, Peter Greven, Venlo, The Netherlands). Tablets were coated with a hydroxypropylmethylcellulose-based film coating system (Opadry^®^ Clear, Colorcon Inc., Indianapolis, IN, USA).

The film-coated tablets were analyzed for appearance, HPLC assay, uniformity of mass, disintegration time, heavy metals, and microbiological quality.

Control is represented by film-coated tablets with the same shape, color, flavor and taste as the intervention tablets.

The supplementation regimen was 2 daily tablets, one before lunch and one before dinner, for 2 months (as for a 60-day continuous integration). Compliance to the supplementation regimen was defined as the number of tablets actually taken by each subject divided by the number of tablets that should have been taken over the course of the study. Adverse events (AEs) were based on spontaneous reporting by subjects as well as open-ended inquiries by members of the research staff. Safety was assessed by laboratory tests performed at baseline and end of treatment (EoT) detailed below, and by recording volunteered adverse events.

### 2.4. Anthropometric Measurements

Nutritional status was assessed using anthropometric measurements at the start of the study, after 30 days and 60 days (that is, at the end of treatment). Body weight and height were measured and body mass index (BMI) was calculated (Kg/m^2^) [28]. Anthropometric parameters were always collected by the same investigator.

### 2.5. Dietary Intake

Participants were asked to maintain their habitual activity and eating behaviors for the duration of the supplementation. A 24 h recall (24h-R) was done and a minimum dietary diversity for women (MDD-W) index was calculated from the data obtained through the 24h-R [29].

The diet was considered diversified if the score was equal to or greater than 5. Only subjects with a score >5 were included in the study. The 24h-R and the MDD-W index were performed at baseline and at the end of the study.

### 2.6. Body Composition

Body composition (FFM, fat mass, and gynoid and android fat distribution) was measured by dual-energy X-ray absorptiometry (DXA) with the use of a Lunar Prodigy DXA (GE Medical Systems). The in vivo coefficients of variants (CVs) were 0.89% and 0.48% for whole body fat (fat mass) and FFM, respectively. Visceral adipose tissue volume was estimated using a constant correction factor (0.94 g/cm^3^). The software automatically places a quadrilateral box, which represents the android region, outlined by the iliac crest and with a superior height equivalent to 20% of the distance from the top of the iliac crest to the base of the skull [30].

### 2.7. Statistical Analysis

The statistical analysis and reporting of this study were conducted in accordance with the consolidated standards of reporting trials (CONSORT) guidelines [31], with the primary analysis based on the full analysis set. Baseline characteristics of the groups were compared using the *t*-test for continuous variables and x^2^ square for categorical variables. Two analysis strategies were subsequently employed. Our main strategy was a multilevel repeated measures analysis with data from the same individuals grouped over three time periods (baseline, 30 days, and 60 days, modelled simultaneously). Time treatment interactions were tested in all models. We considered the effect of the supplementation to vary over the course of the study when the interaction was statistically significant. Clinical and sociodemographic characteristics of both patient and physician were tested in univariate regression models for each of the dependent variables adjusted for gender and age. Those variables that were associated with the dependent variables in these models (*p* < 0.05) were considered statistically significant. As pre-specified, all the analyses were adjusted for age and gender. We used linear regression models to assess differences between groups regarding body composition markers at baseline and 60 days. All analyses were performed by SPSS 21software (IBM, Chicago, IL, USA).

## 3. Results

The baseline clinical parameters of the study are displaced in Table 1. There was a statistically significant difference between the two groups for glycated hemoglobin, apolipoprotein A and B, and Apo B/Apo A. However, these values are within the normal range.

Among the subjects included in the analysis, the mean age was 59.69 ± 7.64 years, and the mean BMI (±SD) was 27.85 ± 2.87 Kg/m^2^.

Table 2 displays the treatment for time effects. Statistically significant interactions were recorded for total cholesterol (*p*-value < 0.009), LDL cholesterol (*p*-value < 0.001), HDL cholesterol (*p*-value < 0.001) and for the total cholesterol/HDL cholesterol ratio (*p*-value < 0.001). No statistically significant effects were recorded for all other variables.

In particular, total cholesterol levels in the supplement group showed a decrease over time of −13 mg/dL (0–60 days). By contrast, the placebo group recorded an increase of +1 mg/dL of total cholesterol. The LDL cholesterol in the supplementation group also showed a decrease over time of −17 mg/dL (0–60 days), whereas the placebo group had a recorded decrease of −1 mg/dL.

HDL cholesterol levels in the supplement group showed an increase over time of 4.39 mg/dL (0–60 days); the placebo group recorded a decrease of −0.9000 mg/dL. The interaction treatment for time was statistically significant.

As shown in Figure 2, the supplementation produced a statistically significant reduction of body weight of −1.194 Kg (CI95%: −1.844; −0.543), whereas in the control group the reduction of −0.383 Kg was not statistically significant (CI95%: −1.055; 0.290). The between-group test showed a non-statistically significant reduction of body weight of −0.811 Kg (*p*-value ns, CI95%: −1.746; 0.124).

The same trend was observed for BMI. In particular, the BMI decrease was statistically significant, at −0.486 Kg/m_2_ (CI95%: −0.762; −0.210) in the treatment group and 0.137 Kg/m^2^ (CI95%: −0.422; 0.149) in the placebo group. The between-group test (supplement minus placebo) showed a non-statistically significant reduction of BMI, at −0.350 Kg (*p*-value ns, CI95%: −0.747; 0.047).

Regarding waist circumference, there was a statistically significant decrease of −1.355 cm (CI95%: −1.919; −0.791) for the supplementation group, while the placebo group showed a non-statistically significant reduction of −0.448 cm (CI95%: −1.031; 0.135). The between-group test (supplement minus placebo) showed a statistically significant reduction of waist circumference of −0.907 cm (CI95%: −1.718; −0.095).

Specifically, regarding the fat mass (FM), the estimated marginal means showed a reduction of −1133 g (CI95%: −1768; −498) in the supplemented group and of −383 g (CI95%: −1039; 272) for the control group. These changes were statistically significant in the supplemented group but not in the control group; the between-group test showed not a statistically significant reduction of −749.900 g (*p*-value ns, CI95%: −1663.160; 163.360).

Another beneficial effect was measured for Visceral Adipose Tissue (VAT), with a statistically significant reduction in the supplementation group of −107.161 g (CI95%: −180.272; −34.050); in the control group there was a slight increase of +1.448 g, but not statistically significant (−74.142; +77.038). The between group test showed a statistically significant reduction of −108.61 g (*p*-value < 0.05, CI95%: −213.772; −3.448).

Regarding safety assessment, laboratory tests performed at baseline and end of treatment did not reveal undesirable concerns, and no relevant adverse effects were recorded. The association of artichoke and Bergamot Phytosome was well tolerated with good compliance.

## 4. Discussion

To our knowledge, this is the first randomized, placebo-controlled study that has demonstrated the efficacy on metabolic profile and body composition of a 2-month artichoke (*Cynara cardunculus* L.) and bergamot (*Citrus Bergamia* Risso) phytosome supplementation in bergamot poor responders with mild hypercholesterolemia. In particular, the present investigation revealed a statistically significant reduction of total cholesterol levels, LDL cholesterol fraction and total/HDL cholesterol ratio, in the supplemented group. In addition, HDL cholesterol showed a statistically significant increase in subjects consuming the supplement.

Our study is in agreement with previous studies that have demonstrated that bergamot is effective in modulating lipid profile, [15,24] in particular when used in combination with other ingredients [14]. More recently, the supplementation with bergamot phytosome, characterized by an improved bioavailability of polyphenols, was shown to be extremely effective in supporting healthy blood lipid levels through the optimization of total cholesterol, LDL cholesterol, HDL cholesterol, triglycerides, as well as serum glucose levels [27]. The mechanism through which bergamot phytosome reduces total and LDL cholesterol is probably linked to the preservation of the natural and unique bouquet of the fruit juice polyphenols fraction [25].

A recent systematic review has investigated the effect of bergamot on lipid profile in humans [32]. Out of twelve studies, nine showed a significant decrease of total cholesterol, triglycerides and LDL cholesterol; one showed a significant decrease in LDL cholesterol only; and two did not find significant changes in any lipid variable. Eight trials reported an HDL cholesterol increase after supplementation with bergamot in any form (isolated phytosterols of bergamot, dry extract of whole bergamot juice and as part of a complex nutraceutical product including other substances such as phytosterols, artichoke extract or vitamin C) [32].

Moreover, a recent meta-analysis of data from nine trials including 702 subjects revealed a significant reduction in both total and LDL cholesterol and triglycerides (respectively of −17.6 mg/dL, −14.9 mg/dL, and −9.2 mg/dL) in subjects with hyperlipidemia using artichoke extract supplementation [33]. No significant improvement in plasma HDL cholesterol concentrations was observed. These results support the lipid-lowering effect of artichoke supplementation in patients with hyperlipidemia [33].

In order to improve the rate of response in bergamot-extract-supplemented subjects with dyslipidemia, the association of bergamot phytosome with a standardized artichoke extract has been developed. The clinical effectiveness of this supplementation, which has been documented with this study, showed surprising benefits. Total cholesterol and LDL cholesterol levels in the supplemented group showed a significant decrease over time of −13 mg/dL and of −1 mg/dL, respectively, and HDL cholesterol levels increased +4.39 mg/dL over time.

Studies have shown that, for each milligram per deciliter (0.0026 mmol/l) reduction in a patient’s LDL cholesterol level, the patient’s relative risk of having a coronary heart disease event is decreased by 1%; therefore, this reduction in LDL cholesterol observed with artichoke (*Cynara cardunculus* L.) and bergamot (*Citrus Bergamia* Risso) phytosome supplementation is likely to be clinically significant as well [34]. Moreover, such changes in cardiovascular risk factors, when applied to the whole population, have significant potential for reducing CVD [35].

Moreover, the increase in HDL of almost 5 mg/dL reported in this study is extremely important in the CVD aspect, since healthy HDL concentrations appear to be more protected against CVD risk [36] for the well-defined role of HDL in reverse cholesterol transport [37], and because HDL has other protective functions, such as anti-thrombotic [38], antioxidant [39] anti-inflammatory [40] and nitric oxide dependent vascular relaxation effects [41].

Considering the other metabolic parameters, such as TG and glycemia, no statistically significant effects were recorded, in accordance with previous studies [27]. However, the TG and glycemic values at baseline were within the normal range.

Another important result of this study concerns the body composition modifications. The 2-month supplementation with the artichoke and bergamot Phytosome supplement led to a statistically significant reduction of body weight and waist circumference. Successful weight loss was confirmed by the reduction of FM and VAT in the supplemented group. This is a relevant result, considering that VAT is associated with metabolic syndrome and cardiovascular disease, and it is also an independent risk factor of all-cause mortality [42]. The management of VAT is a pivotal result, as VAT can now be considered an endocrine organ orchestrating crucial interactions with vital organs and tissues such as the brain, the liver, the skeletal muscle and the heart and blood vessels [43].

In this study, bergamot poor responders with mild hypercholesterolemia showed not only an improvement in lipid profile but also a healthy management of fat storage, as demonstrated by VAT and waist circumference reduction.

It is interesting that the weight reduction and the change in body composition occurred without the subjects following a low-calorie diet. In fact, the patients were asked to maintain their eating habits, which were characterized by a varied diet.

Regarding safety, this study demonstrated that the artichoke and bergamot supplement is safe and well tolerated, with good compliance and no relevant adverse effects.

The main limitations of this study regard the small sample size and the short duration of the intervention, which was limited to two months. Moreover, another limitation regards the enrolled subjects, which included only bergamot poor responders and may limit generalization to the subjects with mild hypercholesterolemia. Finally, a further limitation is due to the fact that it has not been assessed whether patients had changes in the sensation of hunger, a change that could explain the patient weight loss in spite of not following a low-calorie diet.

Therefore, all these findings must be interpreted with caution, and further studies are needed with a larger population size.

## 5. Conclusions

Numerous studies have shown that bergamot extract is effective in reducing hypercholesterolemia, but a percentage of subjects are poor responders and the cause for this is not yet known. As a solution to this limitation, in this study we tested a combination of botanicals (bergamot Phytosome and dry extract of artichoke leaf) with a specific and accurately standardized method of formulation, in subjects with mild hypercholesterolemia and previous evidence of poor response to bergamot. In this study, bergamot poor responders with mild hypercholesterolemia showed an amelioration in their lipid profile with a significant decrease of total and LDL cholesterol over time, while HDL cholesterol increased over time. No significant differences were shown in the placebo group. Moreover, the between-group test showed a statistically significant reduction of waist circumference and of VAT in the supplemented group, even if these subjects did not follow a low-calorie diet.

In conclusion, the synergism between bergamot Phytosome and artichoke dry extract may be a beneficial treatment in subjects who are poor responders to bergamot.

## Figures and Tables

**Figure 1 nutrients-14-00108-f001:**
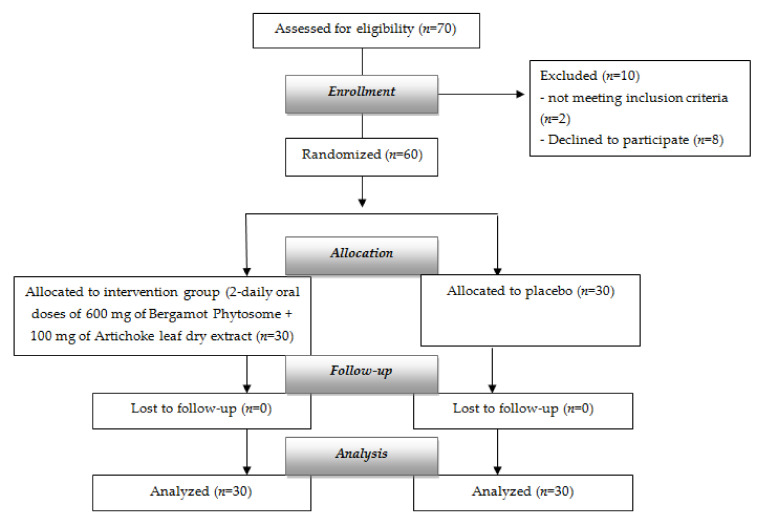
Flow diagram of the study.

**Figure 2 nutrients-14-00108-f002:**
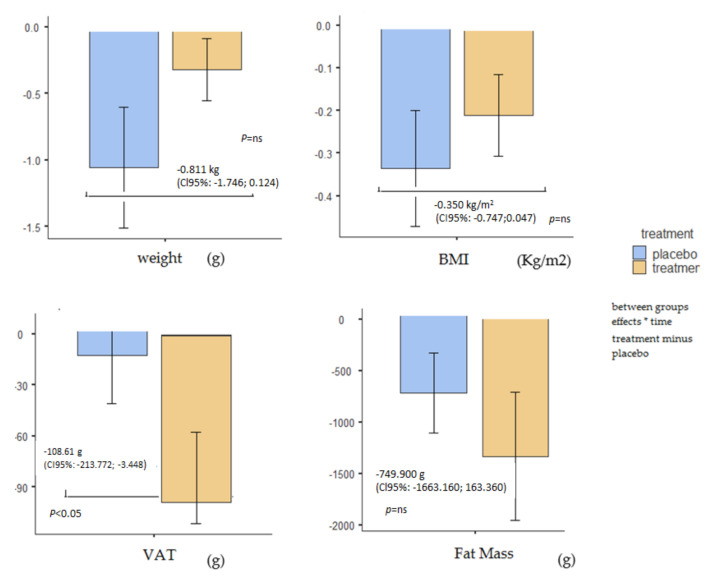
Pre-post body composition measurements in placebo and intervention groups (within and between effects). *: effects for time.

**Table 1 nutrients-14-00108-t001:** Clinical characteristics at baseline.

Outcome	Control Group (*n* = 30) Mean ± SD	Supplemented Group (*n* = 30) Mean ± SD	Total Sample (*n* = 60, Female = 32, Male = 28) Mean ± SD	*p*-Value between Groups
Age (ys)	59.69 ± 7.64	57.64 ± 11.14	58.65 ± 9.54	**0.419**
Weight (Kg)	76.40 ± 18.33	71.49 ± 14.83	73.82 ± 16.62	**0.261**
BMI (Kg/m^2^)	28.38 ± 3.07	27.38 ± 2.63	27.85 ± 2.87	**0.184**
Waist Circumference (cm)	96.63 ± 17.90	94.11 ± 13.00	95.31 ± 15.43	**0.537**
Fat Mass (g)	30,596.54 ± 14,237.25	28,364.42 ± 9271.64	29,423.73 ± 11,835.86	**0.474**
Fat Free Mass (g)	43,506.75 ± 7886.51	43,112.55 ± 9242.26	43,299.63 ± 8554.28	**0.861**
Visceral Adipose Tissue (g)	1100.21 ± 629.85	1108.19 ± 654.05	1104.41 ± 637.15	**0.962**
Total Cholesterol (mg/dL)	234.36 ± 17.18	236.97 ± 24.84	235.73 ± 21.41	**0.640**
LDL Cholesterol (mg/dL)	158.29 ± 19.57	155.83 ± 26.22	157.00 ± 23.14	**0.688**
HDL Cholesterol (mg/dL)	57.18 ± 15.16	59.42 ± 12.60	28.36 ± 13.80	**0.538**
Total Cholesterol/HDL Cholesterol	4.36 ± 1.14	4.14 ± 0.85	4.24 ± 1.00	**0.392**
Triglycerides (mg/dL)	112.36 ± 40.39	108.58 ± 34.91	110.37 ± 37.33	**0.702**
Glycemia (mg/dL)	87.18 ± 6.25	88.19 ± 10.83	87.71 ± 8.89	**0.665**
Insulin (mcIU/mL)	10.66 ± 5.94	10.10 ± 9.08	10.36 ± 7.69	**0.783**
HOMA index	2.34 ± 1.41	2.32 ± 2.44	2.33 ± 2.00	**0.961**
Glycated Hemoglobin (%)	5.95 ± 0.53	5.39 ± 0.34	5.66 ± 0.52	**0.001**
Apolipoprotein A (mg/dL)	159.39 ± 31.17	129.06 ± 25.29	143.46 ± 31.88	**0.001**
Apolipoprotein B (mg/dL)	133.36 ± 18.24	167.45 ± 29.25	151.27 ± 29.87	**0.001**
Apolipoprotein B/Apolipoprotein A	0.88 ± 0.25	1.36 ± 0.42	1.13 ± 0.42	**0.001**
Aspartate Aminotransferase (UI/L)	20.11 ± 5.95	19.65 ± 10.24	19.86 ± 8.41	**0.835**
Alanine Aminotransferase (UI/L)	22.11 ± 8.09	19.23 ± 13.29	20.59 ± 11.13	**0.325**
Gamma Glutamyl Transferase (U/L)	19.46 ± 6.69	21.52 ± 8.74	20.54 ± 7.84	**0.320**
Creatinine (mg/dL)	0.83 ± 0.11	0.82 ± 0.16	0.82 ± 0.14	**0.877**

(*p*-value in bold < 0.05).

**Table 2 nutrients-14-00108-t002:** Estimated marginal means of blood chemistry parameters during the three follow-ups—supplementation for time.

Outcome	Control Group (*n* = 30) Mean	Supplemented Group (*n* = 30) Mean	*p*-Value Between Groups Treatment Time
Total Cholesterol (mg/dL)			**0.009**
T0-baseline	236	237	
T1-30 days	243	228	
T2-60 days	237	224	
LDL Cholesterol (mg/dL)			**0.001**
T0-baseline	157	156	
T1-30 days	165	143	
T2-60 days	156	139	
HDL Cholesterol (mg/dL)			**0.001**
T0-baseline	58.8	59.5	
T1-30 days	58.3	63.6	
T2-60 days	57.9	63.9	
Total Cholesterol/HDL Cholesterol			**0.001**
T0-baseline	4.29	4.13	
T1-30 days	4.5	3.69	
T2-60 days	4.4	3.60	
Triglycerides (mg/dL)			**0.314**
T0-baseline	111	108	
T1-30 days	114	106	
T2-60 days	119	106	
Glycemia (mg/dL)			**0.834**
T0-baseline	87.6	88.2	
T1-30 days	85.7	86.6	
T2-60 days	84.5	86.2	
Insulin (mcIU/mL)			**0.111**
T0-baseline	10.78	10.06	
T1-30 days	10.44	7.22	
T2-60 days	9.98	7.17	
HOMA index			**0.279**
T0-baseline	2.38	2.31	
T1-30 days	2.24	1.58	
T2-60 days	2.32	1.98	
Glycated Hemoglobin (%)			**0.001**
T0-baseline	5.98	5.39	
T1-30 days	5.88	5.4	
T2-60 days	5.73	5.47	
Apolipoprotein A (mg/dL)			**0.423**
T0-baseline	158	129	
T1-30 days	159	127	
T2-60 days	156	130	
Apolipoprotein B (mg/dL)			**0.058**
T0-baseline	133	168	
T1-30 days	137	162	
T2-60 days	133	161	
Aspartate Aminotransferase (UI/L)			**0.862**
T0-baseline	20.1	19.6	
T1-30 days	20.1	20	
T2-60 days	19.5	18.5	
Alanine Aminotransferase (UI/L)			**0.943**
T0-baseline	22.1	19.2	
T1-30 days	22.3	19.7	
T2-60 days	21.7	18.2	
Gamma Glutamyl Transferase (U/L)			**0.416**
T0-baseline	19.4	21.6	
T1-30 days	19.1	23	
T2-60 days	19.8	21	
Creatinine (mg/dL)			**0.995**
T0-baseline	0.827	0.821	
T1-30 days	0.815	0.808	
T2-60 days	0.821	0.813	

(*p*-value in bold < 0.05).

## Data Availability

The data presented in this study are available inside the article.

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
