# Peer review of "Artichoke and Bergamot Phytosome Alliance: A Randomized Double Blind Clinical Trial in Mild Hypercholesterolemia"

_nutrients, 2021, doi:10.3390/nu14010108_

Round 1
Reviewer 1 Report
Manuscript Number: nutrients-1478552, titled:
Artichoke and bergamot phytosome alliance: a randomized double blind clinical trial in mild hypercholesterolemia
the argument is interesting and well treated in the large part of the manuscript with the exception of the introduction section which has to be improved and well argued. The botanical scientific names have to be correctly written. The references section has to be arranged as suggested by Nutrients MDPI. The bibliography has to be improved. The authors have to verify the spacing between words. Inaccuracies in the text.
To the Authors (in detail):
- In the whole manuscript: verify the spacing between words;
- in the whole manuscript: to avoid confusion, please use the correct and updated botanical nomenclature, for example according to www.gbif.org, and also report the authorship and (in brackets) the botanical family at the first mention in the text as follows: Citrus bergamia Risso (Rutaceae). However, you could report the scientific name in the Abstract just like Citrus bergamia. In the rest of the text it is possible to indicate the species as bergamia;
- As for item 2, write the correct complete scientific name of the artichocke you have studied in this work, use the botanical binomial nomenclature;
- Abstract section, lines 19 and 21: bergamot in small letters and Artichoke in capital letters, please, be consistent here and in the whole manuscript;
- Introduction section, line 67, bergamot is not a so known fruit all over the world. Please describe the bergamot fruit, the geographical area of production, the only three existing cultivars. Include some reference describing the shape of this fruit. Support your description with some reference [X1, X2]
[X1] Citrus bergamia, Risso: the peel, the juice and the seed oil of the bergamot fruit of Reggio Calabria (South Italy).
Emirates Journal of Food and Agriculture 32(7) 522-532 (2020).
DOI: 10.9755/ejfa.2020.v32.i7.2128
[X2] The peel essential oil composition of bergamot fruit (Citrus bergamia, Risso) of Reggio Calabria (Italy): a review.
Emirates Journal of Food and Agriculture 32 (11) 835-845 (2020)
doi: 10.9755/ejfa.2020.v32.i11.2197
- Introduction section, line 70, use the binomial botanical nomenclature: Cynara cardunculus in italics;
- Introduction section, line 70, verify the number of spaces before … Cynara cardunculus;
- Lines 74-76: a lot of works have studied the beneficial effects of bergamot fruit on the human health in relation with the cholesterol-lowering activity. Extend, increase and improve your bibliography with [X3, X4, X5]:
[X3] Hypocholesterolaemic activity of 3-hydroxy-3-methyl-glutaryl flavanones enriched fraction from bergamot fruit (Citrus bergamia): In vivo studies.
- Fun. Foods. 7: 558-568 (2014). DOI: 10.9755/ejfa.2020.v32.i7.2128
[X4] Effects of 12-week supplementation of Citrus bergamia extracts-based formulation CitriCholess on cholesterol and body weight in older adults with dyslipidemia: A randomized, double-blind, placebo-controlled trial. Lipids in Health and Disease 16 (1):251 (2017). doi:10.1186/s12944-017-0640-1.
[X5] Atherogenic index reduction and weight loss in metabolic syndrome patients
treated with a novel pectin-enriched formulation of bergamot polyphenols.
Nutrients. 11: 1271 (2019).
- Line 87, Cynara in Capital letters, in addition, Cynara is the genus, please, verify the species for your sentence;
- 1 sub-section, no information is given about the cultivars of bergamot and artichoke, their origin, the number of bergamot fruits and artichocke used for this experiment;
- 1 sub-section: detail how the extracts were prepared;
- Line 98 (standardized), line 133 (analysed): sometime you have used the American spelling and sometime the British one, please, be consistent in the whole manuscript;
- Caption of figure 1 and in the whole manuscript and tables: standardise the spacing before and after ±;
- Line 172, in the table 1 and in the whole manuscript, when you have indicated the significance, sometime you have used p (small letter) and sometime P (capital letter), please, be consistent;
- Line 173, in the caption of table 2 and in the whole manuscript: when you indicate the significance standardise the spacing between symbol and numeric value;
- Figure 2, use the MDPI suggestions to create a figure. In your figure 2 it is not easy to read what it is written in the X and Y axis and inside the figures. Re-arrange the figure;
- Figure 3, use the MDPI suggestions to create a figure. In your figure 2 it is not easy to read what it is written in the X and Y axis and inside the figures. Re-arrange the figures;
- Lines 206-207 and in the whole manuscript: kg in small letters;
- Line 214 and in the whole manuscript: kg in small letter;
- Line 214: m2: 2 as an exponent;
- Line 215: delete one space before … placebo;
- Line 220, verify t after cm;
- Pages 8-9 and in the whole manuscript, always indicate the value of significant or non-significant differences and insert p before the numeric value;
- Line 270, delete one space after … dislypidemia;
- Lines 281-282, verify the spacing between the words;
- References section, please, re-arrange this section in light with the instructions for authors of Nutrients: verify how to write the title of a published paper (for example ref 32), how to write the authors (et al.), and so on;
- Reference 4; verify in the cited paper how it is written the botanical species;
- Reference 21: the scientific names in italics and verify the authors’ names;
- Reference 27; verify in the cited paper how it is written each botanical species;
- Please, write in blue color or evidence differently the corrections you will do.
Regards.

Reviewer 2 Report
Abstract: results portion of the abstract needs to be rewritten for clarity. Are total cholesterol and LDL-C decreased compared to baseline or compared to the placebo group? WC and VAT difference is at what time point, please reword the phrase. What is meant by low-calorie diet? Were all subjects required to maintain an LCD? If not, then how was this assessed, because it puts another variable into the study design.
Introduction: can be shortened and more direct to the point by describing the effect (known) of bergamot and artichoke extract, but the greater details should be moved to the discussion when discussing the results of this study compared to others published. Focus introduction in mentioning the gaps in the literature in regards to hypercholesterolemia, the rationale for this study, and finish with the aims. Otherwise, the introduction becomes more of a discussion. It is not clear the bergamot poor-responders and why authors choose to use it with another extract, again explaining the rationale for this study. Also, please explain the targeted group, are all poor responders mild hypercholesterolemia? Please define mild as this term might be an overreach and why this was chosen.
Methods: How were the bergamot poor responders assessed? Was this via a test of some kind? Please mention and describe it. Please mention figure 1 in section 2.1 and I would also recommend moving it up to give the reader more clarity about the design. Also please expand this section to better explain how the study was carried out. In the abstract, the authors mention about low-calorie diet, but there is no data on dietary assessment mentioned here, please elaborate on that. It is mentioned that the subjects took the supplements with lunch and dinner, is there any additional data, possibly to add to the introduction regarding the bioavailability of these supplements and if they are absorbed better with fats vs water soluble? Please describe placebo and its composition, color, or anything to show readers that participants were blinded.
Results: please explain the randomization system on this trial, it is not clear how posterior analysis was done based on randomization, especially seeing those baseline parameters were significantly different between groups. Ideally, the baseline measures could not have been different in order to assure the quality of the trial. Please elaborate on this. Highly suggest doing an in-house or kit-based measure of lipoprotein functionality. Table 2 is not clear the p-value, if for time for each group, then please show the value for the other group. Please expand on explaining these results a bit better. Table 2 and figure 2 need their own paragraph explaining the results, otherwise, it is too crowded in the body of the manuscript. Figure 2 resolution is very poor, please update. Figure 2 and 3 needs improved legends and descriptions with the units of measures for each parameter in the axis. Please also add * or something is significant between-group or time point and describe in legend.
Discussion: it is mentioned about the effects of bergamot in TG and glucose, were these measured here? Please explain, because it would help compared to previous studies. The mechanism can be more elaborated as there are evidences of phytosterols and polyphenols and lipid metabolism. In addition, there are more limitations to this study such as some type of dietary measurement. Did the subjects report anything? Or self-reported loss of appetite and consequently ate less when taking a supplement? The increase in HDL of almost 5mg/dL is extremely important in the CVD aspect and this should be further explorer here.
Conclusion: how poor responders are detected needs to be clarified in order to make such a conclusion. Otherwise, the study becomes restricted to a certain group type, and how to know what type you are in, needs to be mentioned here. Again, in the introduction, highly recommend describing mild hypercholesterolemia and how that portraits CVD risk with supporting evidence. That way, it will give a stronger rationale for this study.
Overall, the authors did an extremely good job running the trial, it just needs stronger explanation and clarity in the manuscript.
Round 2
Reviewer 1 Report
Manuscript Number: nutrients-1478552, titled:
Artichoke and Bergamot Phytosome alliance: a randomized double blind clinical trial in mild hypercholesterolemia
Review 2 – 21 December 2021
Dear Editor of Nutrients
the argument is interesting and well treated. The authors have included almost all my comments.
Now the only have:
- to write kg in small letters in the whole manuscript, tables and figures;
- to use the italics font in the references section and in the whole manuscript to write the scientific names: Citrus bergamia and Cynara cardunculus, always in italic font.
I suggest to publish this manuscript in the present form after the two above listed corrections.
Regards.

Reviewer 2 Report
Thank you for the improvements and explanation on the topics addressed. The paper is ready for publication.